# The Vaccinia Virus DNA Helicase Structure from Combined Single-Particle Cryo-Electron Microscopy and AlphaFold2 Prediction

**DOI:** 10.3390/v14102206

**Published:** 2022-10-07

**Authors:** Stephanie Hutin, Wai Li Ling, Nicolas Tarbouriech, Guy Schoehn, Clemens Grimm, Utz Fischer, Wim P. Burmeister

**Affiliations:** 1Institut de Biologie Structurale (IBS), Université Grenoble Alpes (UGA), Commissariat à l’Energie Atomique et aux Energies Alternatives (CEA), Centre National de la Recherche Scientifique (CNRS), 38000 Grenoble, France; 2Biozentrum, University of Würzburg, 97070 Würzburg, Germany

**Keywords:** DNA replication, helicase, Pfam domain, poxvirus, cryo-electron microscopy, structure prediction, SF3 helicase, orthopoxvirus, DNA helicase

## Abstract

Poxviruses are large DNA viruses with a linear double-stranded DNA genome circularized at the extremities. The helicase-primase D5, composed of six identical 90 kDa subunits, is required for DNA replication. D5 consists of a primase fragment flexibly attached to the hexameric C-terminal polypeptide (res. 323–785) with confirmed nucleotide hydrolase and DNA-binding activity but an elusive helicase activity. We determined its structure by single-particle cryo-electron microscopy. It displays an AAA+ helicase core flanked by N- and C-terminal domains. Model building was greatly helped by the predicted structure of D5 using AlphaFold2. The 3.9 Å structure of the N-terminal domain forms a well-defined tight ring while the resolution decreases towards the C-terminus, still allowing the fit of the predicted structure. The N-terminal domain is partially present in papillomavirus E1 and polyomavirus LTA helicases, as well as in a bacteriophage NrS-1 helicase domain, which is also closely related to the AAA+ helicase domain of D5. Using the Pfam domain database, a D5_N domain followed by DUF5906 and Pox_D5 domains could be assigned to the cryo-EM structure, providing the first 3D structures for D5_N and Pox_D5 domains. The same domain organization has been identified in a family of putative helicases from large DNA viruses, bacteriophages, and selfish DNA elements.

## 1. Introduction

Orthopoxviruses very recently regained attention with the 2022 outbreak of monkeypox virus (MPXV) infections all over the world with a new mode of human to human transmission [1]. This outbreak was preceded by an increasing occurrence of cases, principally in West Africa and in the Peoples Republic of Congo and occasional exported cases [2]. In the past, monkeypox was a zoonotic disease introduced mainly from rodent reservoirs. The recent expansion is likely to be favored by the waning protection from vaccinia virus vaccination after the end of vaccination campaigns against the now eradicated smallpox virus. So far, there have been little casualties outside Africa, whereas within Africa, the West African and the Congo strains have difficult to estimate mortality rates in the range of 2–7% and 8–13%, respectively [2].

The size of the orthopoxvirus genome varies from 220 kb coding for 223 proteins in cowpox virus to 186 kb for variola virus with 180 open reading frames [3] with a 197 kb genome and about 182 open reading frames in the case of MPXV. There are about 100 shared proteins within the chordopoxvirinae, including the proteins of the DNA replication machinery. As the primase-helicase D5 from vaccinia strain Copenhagen is 99% identical to D5 from monkeypox virus and 98.3% identical to the one from variola virus, this allows the use of the vaccinia virus as safe model system for the study of the shared functions of poxviruses such as DNA replication. At the same time, the development of compounds directed against these proteins should be able to target all orthopoxviruses. Currently, there are two available antiviral compounds against poxviruses, which could not be fully validated in clinical trials: brincidofovir, a chain terminator interfering with viral DNA replication, [4] and tecovirimat [5], targeting the production of extracellular viral particles [6].

The DNA replication machinery consists of five similarly conserved essential proteins: the three proteins that form the DNA polymerase holoenzyme, E9, A20, and D4; the single-stranded DNA-binding protein I3; and the helicase-primase D5. The domain organization of the 95 kDa D5 protein has been analyzed by Hutin and co-workers [7]. We attributed an N-terminal primase domain including also a Cys-cluster domain (res. 1–335), an oligomerization domain requiring residues 323–381, and a helicase domain at the C-terminus (res. 323–785). Oligomerization was required for ATP hydrolysis [8], although D5 lacked a preference for the base of ribonucleotides [9], findings that could also be confirmed for the hexameric C-terminal D5_323–785_ construct [7]. A tight binding of dsDNA and ssDNA could be shown, but helicase activity had remained elusive [7,9].

The C-terminal part of D5 is characterized by the presence of a D5_N motif described in the Pfam database of protein folds [10] (res. 329–470) and a Pox_D5 motif at the C-terminus starting at residue 688, an organization which is widespread for primase-helicases of several representatives of the nucleo-cytoplasmic large DNA viruses (NCLDV), bacteriophages, and helicases from replicating archaeal plasmids. In this family, based on sequence alignments, the presence of D5_N domains (res. 320–464) followed by SF3 helicase domains similar to D5 helicase has been proposed first by Iyer and coworkers [11], who assigned the helicase domain to the AAA+ helicase family characterized by a number of motifs: the Walker A motif comprising the P-loop, the Walker B motif, and Motif C [12].

Based on the presence of a D5_N domain preceding the D5 helicase domain [11], D5 bears some resemblance with the well-characterized helicases from the *E. coli* satellite phage P4 alpha protein [13] and the archaeal plasmid pRN1 [14], all three belonging to helicase superfamily 3 (SF3) from which a helicase activity with a 3′ to 5′ translocation on a single DNA strand had been inferred [13]. Different models for DNA translocation have been proposed [15]. The model systems of SF3 helicases with known X-ray structures are bovine papillomavirus (BPV) E1 [16] and similar adeno-associated virus (AAV) Rep68 [17], polyomavirus large tumor antigen (LTA) helicase [18], and the more distant member porcine circovirus 2 (PCV2) Rep [19]. A structure of the complex of E1 with ssDNA allowed putting forward a model for helicase action where a single strand is translocated through the central pore of a flat helicase hexamer by a “spiral staircase” mechanism [16] involving the cyclic movement of the central β-hairpin loops from an upper ATP-bound state to a lower nucleotide-free state and structure. A “hand over hand“ movement of the subunits along the DNA strand accompanying the cyclic change of the nucleotide-bound state is another variation on this principle [15] and has been proposed for Rep helicase. It involves a staircase-like arrangement of the helicase subunits.

A role of D5 in genome uncoating, a process that liberates the genome after entry into a new host, has been shown where D5 localizes to incoming cores as well as to virus factories and I3-positive pre-replication sites [20]. Additionally, D5 possibly interacts with A20, a part of the DNA polymerase processivity factor, but this interaction is supposed to be rather transient [21]. Given D5’s strict conservation and involvement in genome uncoating and replication, it might be a good antiviral target.

As the hexameric construct of D5 (D5_323–785_) comprising the D5_N domain and the AAA+ helicase domain has already been characterized extensively by small-angle scattering (SAXS) and negative stain electron microscopy [7] but did not yield crystals diffracting to an exploitable resolution, we took advantage of the recent development of cryo-electron microscopy (cryo-EM) in order to determine its high-resolution structure.

## 2. Materials and Methods

### 2.1. Protein Expression and Purification

Full-length D5 (D5fl) has been expressed and purified as described in [7]. Two purification steps have been added: a glycerol gradient purification and an ion exchange purification on an UNO Q column (Bio-Rad, Marnes-La-Coquette, France). The glycerol gradient used 10–30% glycerol in 10 mM Tris-HCl pH 7.0, 150 mM NaCl, 1 mM DTT centrifuged for 25 h at 30,000 rpm in a SW32 Rotor at 4 °C. The fractions containing D5 hexamer were pooled, diluted to a concentration of 25 mM NaCl in buffer without salt and glycerol, and loaded on the UNO Q column (Bio-Rad). The protein was eluted with a NaCl gradient up to 500 mM NaCl in the same buffer, and peak fractions were pooled and used directly for negative stain EM.

The 53.4 kDa D5_323–785_ fragment was cloned as a His-tagged tobacco etch virus (TEV) protease-cleavable construct using the pProEx HTb vector with the primers 5′-gcgccatggg taataaactg tttaatattg cac-3′ and 5′-atgcaagctt ttacggagat gaaatatcct ctatga-3′, resulting in two additional N-terminal residues Ala-Met from the TEV cleavage site. It was produced and expressed as described in [7], but the last purification step was replaced by an ion exchange chromatography step using an Enrich Q 5 × 50 column (Biorad). The protein was loaded in 20 mM Tris-HCl pH 7.0, 6.67 mM NaCl, 13.32 mM KCl, 2 mM MgCl_2_, and 1 mM DTT, and the protein was eluted with a gradient ending in 20 mM Tris-HCl pH 7.0, 166 mM NaCl, 334 mM KCl, 2 mM MgCl_2_, and 1 mM DTT. Fractions of the second peak were pooled, and the buffer was exchanged to buffer K containing 5% glycerol, 50 mM Tris-HCl pH 7.0, 50 mM NaCl, 100 mM KCl, 2 mM MgCl_2_, and 10 mM β-mercaptoethanol by 2 concentrations to 100 µL in a 2 mL centrifugal concentrator (30 kDa cut-off, Millipore, Burlington, MA, USA).

### 2.2. Negative Stain EM

To the sample (0.08 mg·mL^−1^ D5fl) were added an additional 10 mM MgCl_2_, 1 mM ADP, and 2 mM BeF_3_. About 4 µL of a D5fl sample were applied to a mica sheet covered with a film of evaporated carbon. The carbon film was floated off the mica in glycerol-free buffer, retrieved, and placed onto a 400-mesh copper electron microscopy grid. The sample was subsequently stained with 4 µL of 2% sodium silicotungstate (SST), blotted, and dried. Imaging was performed on an FEI F20 transmission electron microscope operating at 200 kV. Images were recorded on a 4k × 4k Eagle digital camera with a pixel size of 2.17 Å.

### 2.3. Cryo-EM Sample Preparation

A complex with an end-labelled dsDNA 30mer was used. An oligonucleotide with the sequence biotin-5′-ccgaatcaggaagataacagcggtttagcc-3′-digoxigenin (DIG) was annealed to an unlabeled oligomer 5′-ggctaaaccg ctgttatctt cctgattcgg-3′ (Eurofins, Ebersberg, Germany) after heating the mix to 95 °C for 5 min and overnight slow cooling to room temperature in a beaker. Then, 7.5 µM D5_323–785_ hexamer was mixed with 9.2 µM of dsDNA oligomer in a volume of 6.5 µL. After 10 min incubation, samples were diluted 10 times in buffer K without glycerol. Next, 4 µL samples were deposited on glow discharged R1.2/1.3 holey carbon grids (Quantifoil Micro Tools GmbH, Großlöbichau, Germany) and blotted for 2.5 s at blot force 1 with 0.5 s drain time using a Vitrobot Mark 4 (Thermo Fisher Scientific, Illkirch-Graffenstaden, France) prior to flash-freezing in liquid ethane.

### 2.4. Cryo-EM Data Acquisition Parameters

**The parameters** are described in Table 1.

### 2.5. Three-Dimensional Reconstruction

Cryo-EM Data treatment used Relion 3.1 [22,23] in a standard way, as described in the manual [24], running on a PC with 2 Intel Xeon processors with 80 threads and 4 Nvidia GeForce GPUs under Linux Ubuntu 20. Movies were aligned with MotionCorr, and contrast transfer functions were calculated with CtfFind4 [25]. Initially, particles were picked with the model-free “Laplacian of Gaussian” algorithm on micrographs with a resolution better than 3.6 Å. Appendix A shows the pipeline for particle reconstruction. After first 2D classifications, good classes were used for particle picking with 2D templates. Unsupervised 3D reconstruction in C1 gave a first model, which was symmetrized in C6 and used for 3D classifications. The best class yielded a 4.1 Å resolution structure after refinement, polishing and post-processing steps. The structure was used in a new round of particle picking with 3D templates using all micrographs in order to maximize the number of available particles for reconstruction in C1. After several rounds of 2D classifications, unsupervised reconstruction, 3D classification, and refinement (Appendix A), a 6.6 Å structure was obtained. Low-resolution reconstructions of D5fl in negative stain used Relion 3.1 and a similar protocol. Classifications did not use CTF-correction up to the first peak.

### 2.6. AlphaFold2 Prediction

The amino acid sequence of vaccinia virus (Copenhagen strain) D5 was submitted to a Colab notebook running a simplified version of Alphafold v2.1.0 (https://colab.research.google.com/github/deepmind/alphafold/blob/main/notebooks/AlphaFold.ipynb accessed on 27 July 2022).

### 2.7. Model Building and Refinement

The Alphafold2 model of D5 corresponding to the D5_323–785_ construct was split into three domains according to the observed domain boundaries: collar domain (res. 323–403), AAA+ helicase domain (res. 404–705), and C-terminal domain (res. 706–785). The domains were fitted as rigid bodies into the cryo-EM electron density using Chimera [26]. Manual rebuilding used Coot [27]. Real-space refinement and structure validation [28] were carried out using Phenix [29]. The Phenix real-space refinement using global minimization and B-factor refinement (adp) is able to handle resolutions up to 6 Å [30]. We opted for a conservative refinement protocol using Ramachandran, secondary structure, and ncs constraints for both structures. The software installation at IBS used SBGrid [31].

## 3. Results

### 3.1. Domain Structure of D5

Exploratory EM imaging of full-length D5 (D5fl) in negative stain under different conditions regarding the presence of Mg^2+^, ATP, ADP, or ADP, and BeF_3_ showed the flexible attachment (Appendix A) of the N-terminal oligopeptide to the hexameric D5_323–785_ [7]. As the ribonucleotide hydrolase activity of D5_323–785_ had also been indistinguishable from D5fl [7], we used the D5_323–785_ construct for high-resolution work, assisted by an Alphafold2 prediction, which yielded a structure of D5 (Figure 1a) with a high reliability index (Figure 1b) and allowed us to assign five domains: the primase domain and a distinct Zn-binding (or Cys-cluster) domain in the N-terminal polypeptide not present in D5_323–785_ used for cryo-EM reconstruction, a collar domain, an AAA+ helicase domain, and a C-terminal domain.

Cryo-EM of D5_323–785_ yielded a variety of projections (Appendix A) that generated an electron density map with a nominal resolution of 4.1 Å based on the half-set Fourier shell correlation (Appendix A) when six-fold symmetry was inferred. Data processing in lower symmetry point groups (C3 or C1) led to significantly lower resolutions between 6 and 7 Å. Reconstructions appeared to be isotropic when analyzed by 3DFSC [34] and showed the expected behavior in a ResLog plot [35] (Appendix A). The electron density showed an organization in three domains, which could be fitted with the Alphafold2 models of the corresponding domains (Figure 2a,b). The density corresponding to the collar domain is defined best with estimated local resolutions (Figure 3a) of 3.9 to 4.5 Å (a sample of the density is shown in Figure 3b). The one with the AAA+ helicase domain has a lower resolution (4–6 Å, example density in Figure 3c), and the resolution of the C-terminal domain is in the 6 Å range. The electron density of the C-terminal domain was also weaker, requiring a lower contour level (Figure 2b). A few residues were manually rebuilt in order to fit the experimental electron density before the model was submitted to real space refinement (Table 2). Only a little helix of the AAA+ domain from the Alphafold2 model (res. 632–644, Figure 1c, arrow) could not be located and has been omitted from the final model together with the last three residues of the C-terminal domain. Adjustments in the collar domain were minimal and remained very limited in the AAA+ helicase domain. Due to the low resolution of the C-terminal domain, the final model stayed as close as possible to the Alphafold2 model. The refined structure of D5_323–785_ conserves largely the geometrical quality of the initial Alphafold2 model (Table 2, third column), which it matches very well (Figure 1c) with an rms deviation of 1.3 Å for all atoms. Individual domains superpose with 1.3 Å rms for the best-defined collar domain and 1.0 Å rms for the other domains, values which are certainly influenced by a strong bias towards the Alphafold2 model in the less well-defined parts of the structure. The hexamer has a large central channel, which could be large enough to accommodate dsDNA, of which the diameter is indicated (Figure 3d, green circle).

Comparing the position of the three domains of D5_323–785_ with the Pfam [10] analysis of D5 (Figure 4a), it became clear that the collar domain is part of the Pfam D5_N domain, which extends further into the AAA+ helicase domain (Figure 4b,c). The D5_N domain is composed of an N-terminal α-helix, followed by a three-stranded β-sheet, a bundle of three more helices, another three-stranded β-sheet, and an alternation of two short helices with extended backbone conformations ending with an α-helix, which marks the transition into the helicase domain (Figure 4a). The C-terminal domain corresponds to a Pox_D5 domain. It was tempting to assign the remaining part of the AAA+ helicase domain to the Domain of Unknown Function 5906 (DUF5906) in Pfam, present very often between D5_N and Pox_D5 domains, although it had not been assigned for poxvirus. A Dali search [36] with the AAA+ helicase domain for proteins with similar folds (Table 3) confirmed the assignment of the AAA+ helicase domain, as it is very similar to the one of deep-sea vent phage NrS-1 polymerase [37] containing a DUF5906 domain.

The NrS-1 polymerase-helicase is composed of an N-terminal Primpol domain acting as the DNA primase and polymerase [38] and a hexameric C-terminal domain with a likely, but unproven, helicase activity. The similarity of the C-terminal domain of NrS-1 polymerase-helicase with D5_323–785_ extends beyond the DUF5906 domain and covers the entire D5_323–785_ fragment (Figure 5a). For example, the little β-sheet (res. 339-357, Figure 3b) of the D5_N domain is also present with the same orientation relative to the helicase domain (Figure 5b). Since in addition to the three β-strands only the helix at the N-terminus of the domain is conserved, the similarity has not been detected by the Dali search against the PDB database. The structural similarity also concerns the Pox_D5 domain, where some secondary structure elements can be matched in space, and furthermore, their topology is conserved (Figure 5a).

D5_323–785_ is the first structural representative of the Pfam domains D5_N and Pox_D5, while the DUF5906 domain has already been described for the NrS-1 helicase. Moreover, at the level of the structure, the NrS-1 helicase is the most similar protein of known structure compared to D5_323–785_.

### 3.2. DNA Binding of D5_323–785_

The hexamer shows a central channel with a minimal diameter of about 16 Å at three points of constriction: the level of the basic cluster in the collar domain (21 Å), the β-hairpin loop (16 Å), and the DNA binding loop in the AAA+ helicase domain (18 Å). The channel would just be sufficiently wide for the passage of dsDNA. The lysine residues of the basic cluster and the loops inside the AAA+ helicase domain are flexible and correspond to the least well described parts of the structure, so that the passage of dsDNA cannot be excluded. The sample used for the cryo-EM data collection contained a roughly equimolar complex of D5_323–785_ hexamer with a dsDNA 30-mer obtained by mixing. The dsDNA oligomer had been used previously for the characterization of DNA-binding by D5_323–785_, where a 1:1 complex had been observed [7]. A blunt-end dsDNA construct had been chosen due to the proposed presence of dsDNA inside a polyoma virus LTA helicase hexamer structure retracted since then [39].

In 2D classifications, additional electron density was visible above the collar domain in the center of the hexamer (Figure 6a, green arrow). Furthermore, electron density for the C-terminal domains on the bottom of the hexamer appeared to be asymmetric (Figure 6a, pink arrow). At the N-terminal side, a symmetry mismatch must occur between the dsDNA and the six-fold symmetric collar domain. These features indicate a breakdown of the six-fold symmetry used so far. In consequence, a reconstruction without imposed symmetry has been calculated, leading to a resolution of 6.6 Å (Appendix A), significantly lower than the reconstruction in C6 as expected from the reduced data redundancy. The electron density of D5_323–785_ reconstructed without symmetry (Figure 6b,c) showed disorder at the level of the C-terminal domains, with two domains being well defined, two domains showing weak electron density, and two domains being invisible (Figure 6d,e) in agreement with the results of negative stain EM on D5fl, which showed also the disorder of the C-terminal domain (Figure 6g), whereas collar and AAA+ domain matched the density very well.

Above the entrance to the central channel of the collar domain, additional electron density is visible (Figure 6b), which ends in the central channel at the level of the basic cluster. Additional density starting at the level of the β-hairpin loops (dotted line, Figure 6e) is present in the middle of the central channel of the DUF5906 helicase domain (Figure 6c). It is not defined well enough to assign single-stranded or double-stranded DNA strands. The additional electron density on top of the collar domain has been tentatively interpreted as six base pairs of dsDNA with two additional bases extending into the central channel and two bases running on the top surface of the hexamer (Figure 6d–f), even though the electron density is not well ordered. The electron density of the protein was interpreted starting from the model refined in C6, where two of the C-terminal domains have been omitted. The model could be refined in real space using Phenix (Table 2). The disorder of the DNA is not surprising, as the classification of the particles will be dominated by the differences in the C-terminal domains unless there is some cross-talk between their conformation and the orientation of the bound DNA.

## 4. Discussion

### 4.1. Domain Structure of D5

The structure shows the power of structure calculation by Alphafold2, furthermore, unexpectedly, Alphafold2 predicted the relative orientation of the three domains in absence of any information about the hexameric organization of D5_323–785_ (Figure 1c). This suggests that the interface contacts between the domains are sufficient in order to define their relative orientations.

The structure of D5_323–785_ containing a D5_N domain followed by DUF5906 and Pox_D5 is widespread in viral helicases, but so far, experimental structural information was missing. Our structure gives the first view on the high-resolution structure of these domains and their arrangement. The limits of the sequence-based domain assignment used for the definition of the D5_N domain become obvious. Only the part corresponding to the collar domain is an independent structural domain, whereas the other part forms one unit with the helicase domain so that D5_N domains probably do not exist in absence of a following DUF5906 domain (Figure 4c).

A Dali search with the collar domain surprisingly yielded the ribosomal protein S17 [40] as a best hit, even before the one for papillomavirus E1 protein (Table 3). A closer inspection of the structural alignment with S17 (Figure 4c) and the E1 collar domain (Figure 4d) showed that despite the exact superposition of the helical structures, the two proteins do not share the three-stranded β-sheet present in D5, indicating an early divergence in evolution between the family of D5-related helicases and E1-related helicases. Surprisingly, the D5_N domain of NrS-1 helicase and Rep68 have not been detected by Dali, although they share one helix and the three-stranded β-sheet with D5 (Figure 5b). Lower Dali scores were obtained with polyoma virus LTA and several subunits of MCM helicases (Table 3). LTA acquired an additional Zn-binding module (not shown) in its collar domain. On MCM helicase subunits, the domain corresponding to the collar domain is located further away from the central channel of the helicase hexamer than for E1 and LTA (not shown). The absence of the three-stranded β-sheet being part of the D5_N domain explains the low Dali scores. The surprising similarity with the ribosomal protein S17 (Figure 5c) suggests that a helicase similar to E1, LTA, or MCM might have gone moonlighting as an additional subunit of the ribosome followed by a potential degeneration of its C-terminal part to a largely unstructured extension running along the ribosomal surface.

Dali homologies with the C-terminal domain appear weak with a best, but not necessarily significant, hit with the DNA-binding domain of the human RFX1 winged-helix transcription factor ([41], pdb entry 1dp7). In particular, the orientation of the helix inserting into the major groove of the cognate DNA is not conserved (not shown).

### 4.2. The AAA+ Helicase Domain

The best hit of the Dali search with the AAA+ helicase domain was the helicase domain of bacteriophage NrS-1. This similarity extends over all its structure, which could be matched with D5_323–785_ in a structural alignment with FATCAT ([42], Figure 5a) confirming the similar domain organization as suggested by Pfam. This positions D5_323–785_ within a clade together with helicases of giant DNA viruses, of P4-related bacteriophages comprising NrS-1, and of replicating plasmids. These helicases diverged early from other SF3 helicases such as E1, Rep68, and LTA. To date, the systematic prediction of protein structures of bacterial genomes by Alphafold2 and accessed on 17 June 2022 through https://pfam.xfam.org yielded three structures containing D5_N domains, which could also be aligned structurally with D5 and thus belong to the same clade: the structures of a probable phage protein from *M. tuberculosis* (O06608) and two phage-related proteins from *K. pneumoniae* (A0A0H3GIU0 and A0A0H3GLV7).

The different sequence motifs of AAA+ helicases, namely, Walker A, Walker B, Motif C, and the arginine finger, could be assigned for D5 [12] (Figure 7a). The role of the key residues of the motifs had been confirmed previously by site directed mutagenesis [8], and a temperature-sensitive mutant Pro682Ser inside D5_323–785_ had been described. The observed position of the proline residue capping an α-helix upstream of a stretch of residues running on the surface of the hexamer (Figure 7a) would explain the observed lower expression of the temperature-sensitive mutant due to inefficient folding, even at permissive temperature.

In our structure, any additional electron density, which could be interpreted as nucleotide, was missing so that no information about the nucleotide-binding site could be obtained. This is not surprising, as the sample contained D5_323–785_ with a 30mer dsDNA in absence of additional nucleotide. No information from a closely related protein could be inferred, as the structure of NrS-1 helicase has also been determined in a nucleotide-free state. A structural alignment of the Walker A motifs comprising the P-loop, Walker B motif, and Motif C allows the superposition of NrS-1 helicase, E1, and LTA nucleotide binding sites and the ADP molecules bound to E1 and LTA (pdb entries 2gxa and 1svl, [16,18]). However, serious clashes (marked with green stars in Figure 7c,d) would occur when the radically different positions of the base of the ADP molecules in E1 and LTA structures are used. This suggests that in D5 the recognition of the base probably differs from the modes observed for E1 and LTA or that the structural elements involved in base recognition are flexible and the binding site is only structured in presence of a nucleotide. A difference in the binding mode would not be surprising, as D5 appears to be promiscuous regarding the base at the level of the basal nucleotide hydrolase activity [7]. Interestingly, the positions of the nucleotide base of E1 and LTA would also clash with the structure of NrS-1 (not shown) speaking in favor of a possible common nucleotide binding mode, but as at the same time, the mobile and functionally important part of the structure between the β-hairpin loop, the DNA-binding loop, and the Walker B motif diverges between D5 and NrS-1 helicase (Figure 7b), so extreme precaution is required in attempts to infer functional similarities.

D5 shares with Nrs-1 helicase, E1, the hexameric structure of Rep68, and LTA a planar arrangement of the collar and AAA+ domain subunits, which are in turn in close contact in contrast to the more remote PCV2 Rep protein, where the subunits show a staircase arrangement and the collar and AAA+ domain are connected by a loose linker. For E1 helicase, a helical staircase mechanism of DNA translocation has been proposed, where the β-hairpin loops play a central role as they descend (if the collar domain is placed at the top) progressively from the initial ATP-bound state over an ADP-bound state to a final nucleotide-free state. When the β-hairpin has reached its bottom position, it disengages from the DNA. Conformational changes induced by the bound nucleotide are transmitted to the β-hairpin and the DNA binding loops, whereas positional changes of the arginine finger trigger nucleotide binding and ATP hydrolysis of the neighboring subunit. So far, it is assumed, but not proven, that D5 uses a similar mechanism shared within the above mentioned SF3 helicases but different from the one of PCV2 Rep [19]. Regarding the structure of the β-hairpin loops, D5 is similar to E1 and LTA (not shown), in contrast, the DNA-binding loops of E1 are much shorter. Residue conservation of these loops with central role for function [43] is very low when experimental or predicted structures of several related helicases are aligned (Figure 7e).

Whereas, globally, the model from Alphafold2 agreed very well with the experimental electron density, the refined model only deviates for the more flexible parts of the helicase domain (Figure 6a) from the Alphafold2 prediction. The corresponding structural elements are located on one side of the central β-sheet and form a metastable structure with important conformational changes during the different states of DNA translocation, as described for E1 helicase [16,44], easily explaining the difficulty of a reliable structure prediction adding to a less reliable experimental structure because of the limited resolution.

### 4.3. DNA Binding of D5_323–785_

The central channel of D5_323–785_ would be just wide enough to accommodate dsDNA (Figure 3d) as the residues constricting the channel are flexible or situated in the flexible DNA-binding and β-hairpin loops. Moreover, the cryo-EM structure shows that D5_323–785_ is able to bind dsDNA oligomers at the entrance to the central channel, even though it has not been possible to demonstrate an helicase activity using different forked substrates ([8] and unpublished results). The observed positioning of the DNA would be compatible with a strand separation occurring at the entrance of the D5_323–785_ hexamer with one strand running on the top surface of the collar domain and one strand running inside the central channel, as described recently for E1 [45] (Figure 6d–f). It is likely that the DNA backbone interacts with a cluster of basic residues at position 387–390, which is conserved throughout the orthopoxviruses. The dsDNA could interact with the Arg387 residues of several subunits, whereas Lys388 would rather interact with the excluded strand and Lys390 with the translocated strand (Figure 6f) but a continuation of the electron density inside the collar domain is clearly not seen. The two residues resulting from the cloning, Ala321 and Met322, are located next to the density of the DNA and may also interact with it so that it cannot be excluded that the N-terminal truncation of D5_323–785_ affects the interaction with DNA.

With the limited quality of the electron density, we prefer not to hypothesize about the presence of ssDNA or dsDNA in the central channel of D5_323–785_ explaining the observed electron density. The flexibility of the C-terminal domains already suggested from negative stain EM (Figure 6g) may reflect a role in the interaction with a DNA strand exiting from the central channel, but the different degrees of order of the domains could also be an artifact resulting from an asymmetric adsorption of the particles on the air–liquid interface during cryo-EM sample preparation as one projection with a slightly offset six-fold axis is dominant (Appendix A). The high particle density observed in the micrographs (Appendix A, detail of a micrograph) despite the low sample concentration also speaks in favor of such an adsorption.

## 5. Conclusions

Most aspects of the helicase action of D5 still remain enigmatic but the available structure will help greatly to orient future experiments required to establish the mechanisms of helicase loading and translocation. The combination of cryo-EM and Alphafold2 structural calculations has been proven to be very powerful for structure determination even in the context of oligomeric proteins, which are not fully implemented into the Alphafold2 algorithm yet. Our structure indirectly validates Alphafold2 calculations of thousands of viral helicase structures related to D5, which are waiting to be released.

## Figures and Tables

**Figure 1 viruses-14-02206-f001:**
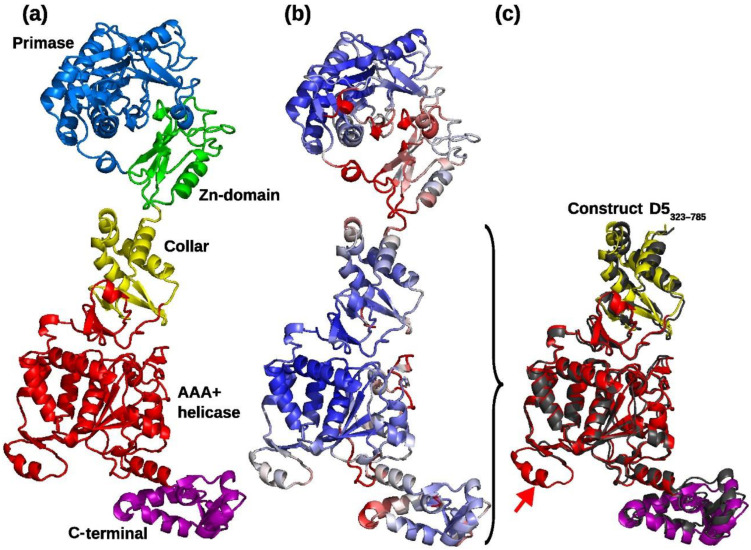
Alphafold2 model of the helicase primase D5. (**a**) Domain organization: Primase, blue; Zn-domain, green; collar domain, yellow; AAA+ helicase domain, red; C-terminal domain: purple. (**b**) Local-distance difference test (LDDT) reliability index from Alphafold2 [32] ranging from 70% (red) to 100% (blue). (**c**) Superposition of the Alphafold2 structure colored as in panel (**a**) onto the refined D5_323–785_ structure colored in gray. An arrow marks the predicted helix missing in the density. Figure prepared with with PyMol [33].

**Figure 2 viruses-14-02206-f002:**
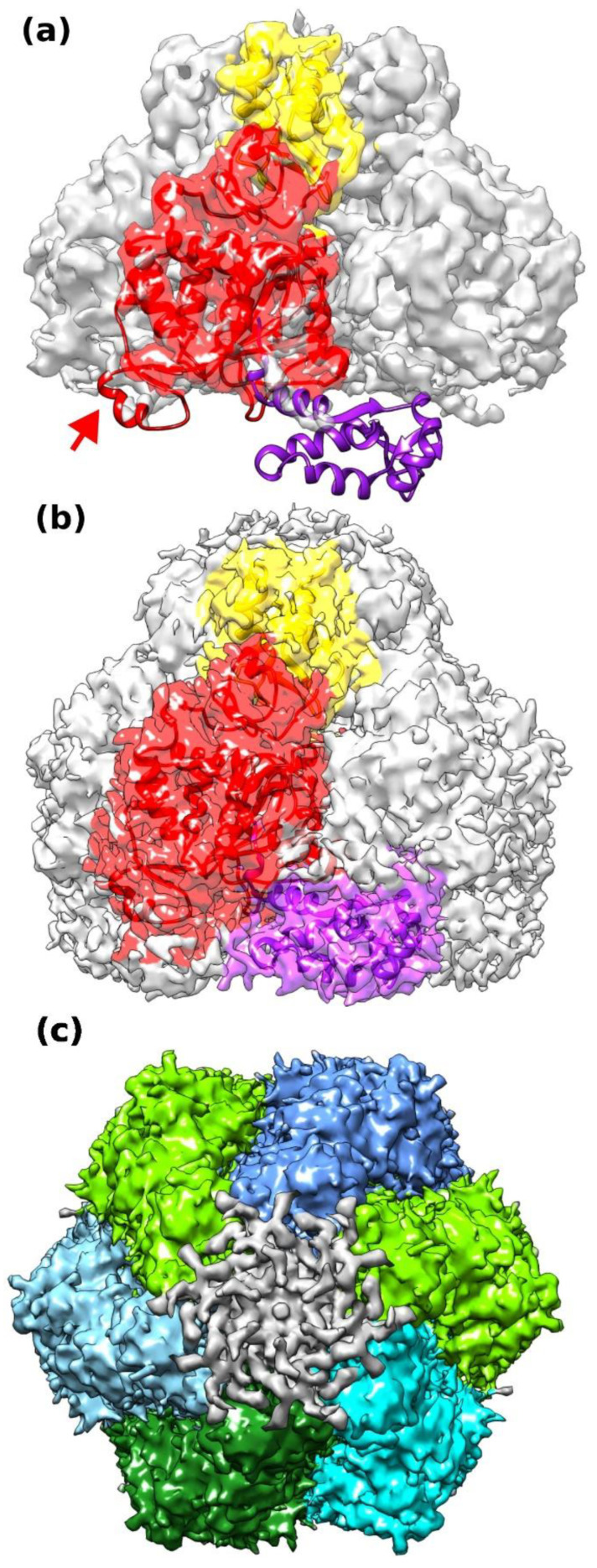
Electron density of D5_323–785_ reconstructed using strict C6 symmetry and its interpretation. (**a**) Electron density contoured at high contour level with a docked collar domain (yellow), a docked AAA+ helicase domain (red), and the C-terminal domain (purple) of the Alphafold2 model. Docking of the C-terminal domain required a lower contour level as shown in panel (**c**). The electron density is colored according to the domains. (**b**) At lower contour level, electron density for the C-terminal domain appears and allows the rigid body fit of the Alphafold2 model. (**c**) The assembly of the 6 subunits in the electron density contoured at low level as in panel (**b**) and colored by subunit. Rotationally averaged DNA density at the entrance of the collar domain is shown in grey. Illustration made with Chimera.

**Figure 3 viruses-14-02206-f003:**
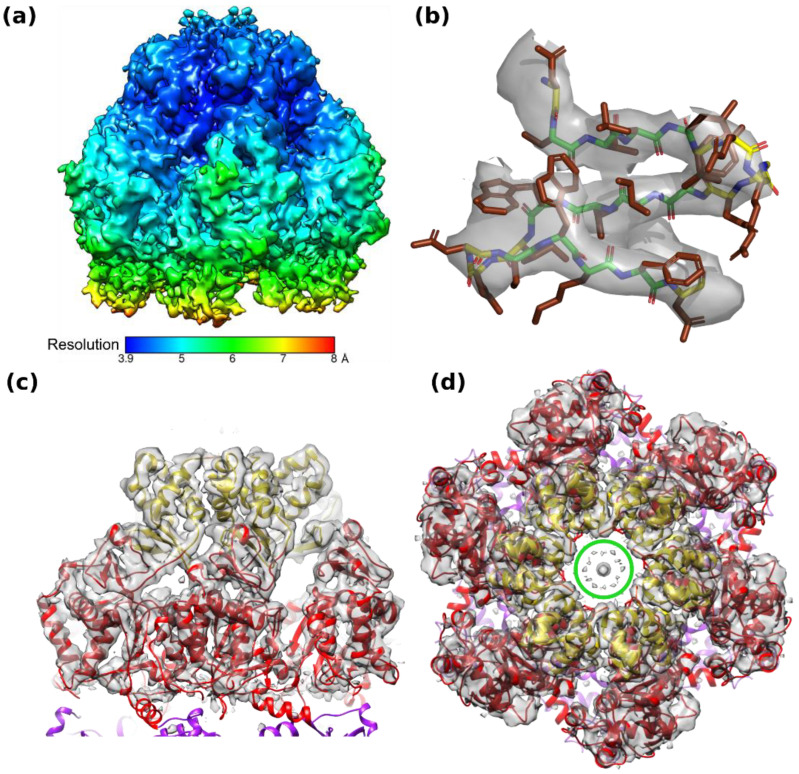
Electron density and resolution of the C6 reconstruction. (**a**) Local resolution of the final map calculated in Relion. (**b**) Sample electron density for the small β-sheet of the collar domain characteristic of D5 (res. 339–357, green carbon atoms in the main chain). Loops are colored with yellow carbon atoms. Side chains are shown with brown carbon atoms. Panel prepared with PyMol. (**c**) Side view of the hexamer with electron density colored at high contour level showing the fit of the AAA+ helicase domain. Colors as in Figure 2. (**d**) Top view showing the central channel. The circle indicates the diameter of dsDNA.

**Figure 4 viruses-14-02206-f004:**
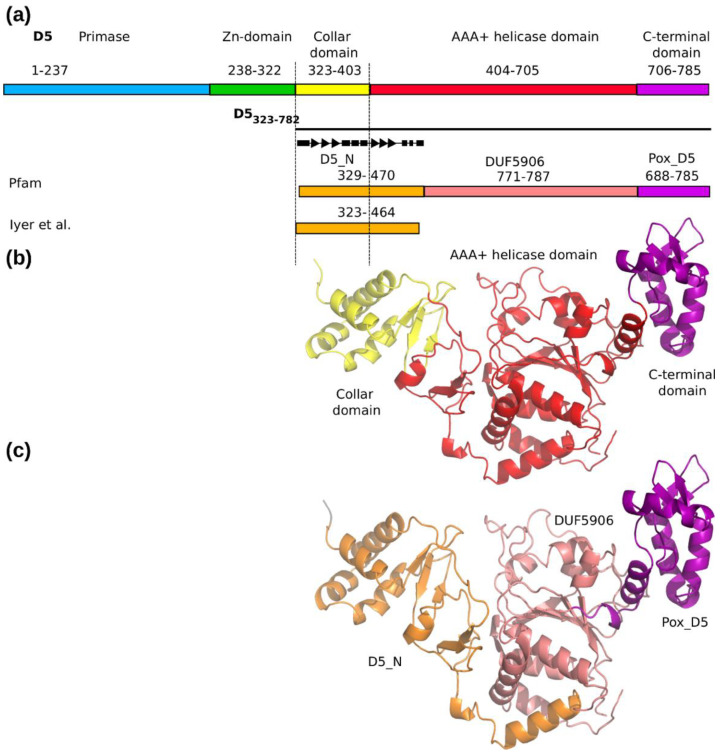
Domain structure of D5. (**a**) Domain structure of D5 based on the cryo-EM structure and Alphafold2 prediction for the N-terminal part compared to the domain structure of D5_323–785_ using sequence-based Pfam domain definitions and finally the attribution of the D5_N domain by Iyer and coworkers [11]. The topology of the D5_N domain is symbolized by triangles for β-strands and rectangles for α-helices. (**b**) Domain structure of D5_323–785_ based on the cryo-EM structure determined in C6. Colors as in panel (**a**). (**c**) The sequence-based Pfam domain structure mapped onto the cryo-EM structure. Colors as in panel (**a**). Illustration made with PyMol and Inkscape (https://inkscape.org/ accessed on 6 October 2022).

**Figure 5 viruses-14-02206-f005:**
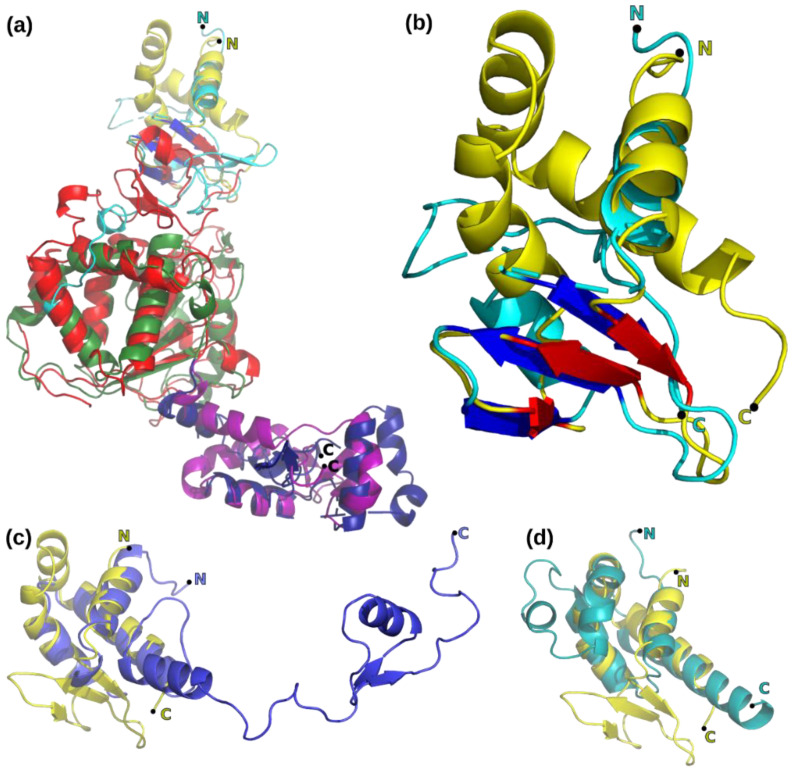
Structures related to D5_323–785_. N and C-terminal extremities of the fragments are labeled. (**a**) Flexible structural alignment using the FATCAT server of the NrS-1 helicase domain (pdb entry 6k9c) with D5_323–785_ (domains colored as in Figure 1, the 3 β-strands in the collar domain are colored in red). Cyan, green, and blue are used respectively to color the NrS-1 helicase with the 3 β-strands shown in blue. The FATCAT alignment creates chain breaks for the molecule submitted to the alignment. (**b**) Zoom on the superposition of the collar domains of panel (**a**). (**c**) Similarity between the collar domain of D5 (yellow) and a part of the ribosomal protein S17 (pdb entry 6zxh, blue). (**d**) Similarity between the collar domains of D5 (yellow) and papillomavirus E1 helicase (pdb entry 2gxa, turquoise). Illustration made with PyMol.

**Figure 6 viruses-14-02206-f006:**
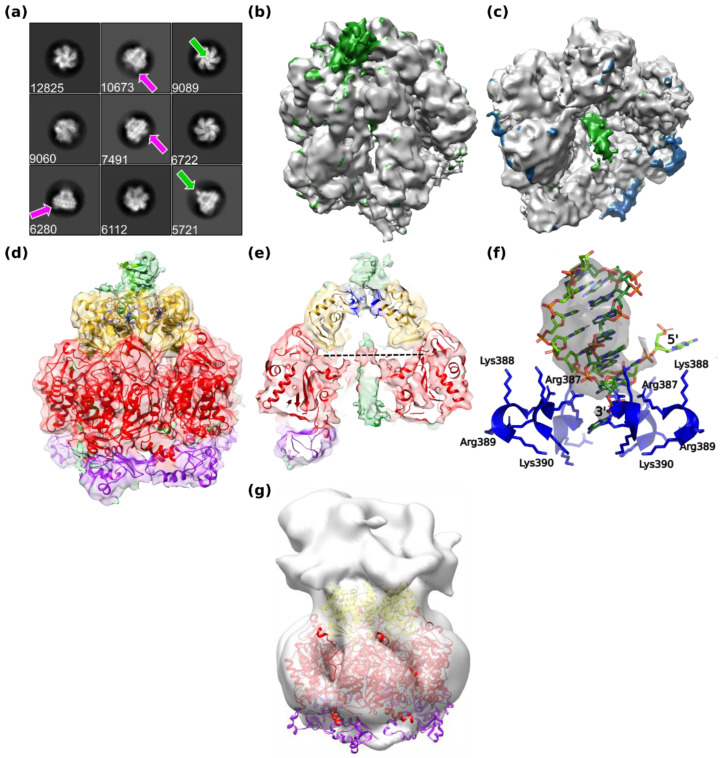
Structure calculation without imposed symmetry. The corresponding flowchart is given in Appendix A. (**a**) Two-dimensional classes used for reconstruction and their numbers of particles. Density corresponding to DNA is indicated by green arrows and the heterogeneity at the level of the C-terminal domains is highlighted by pink arrows. (**b**) Electron density of the 3D reconstruction obtained after further rounds of classifications and 3D refinement using 26710 particles. Additional density shown in green is visible at the entrance of the collar domain that is not explained by a fitted model of the D5_323–785_ hexamer. (**c**) Slice of the hexamer with additional density close to the 6-fold axis shown in green, which is not explained by a fitted model of D5_323–785_. Other unmodeled density (in blue) corresponds to disordered C-terminal domains. (**d**) Model and density of the hexamer colored by domains with modelled dsDNA at the entrance of the collar. (**e**) Slice through the electron density colored according to the domains of the refined D5_323–785_ hexamer including dsDNA showing the additional electron density along the central channel (colored in green). The residues of the basic cluster of res. 387–390 are shown in stick representation (blue carbon atoms). The electron density attributed to dsDNA ends at the level of the basic cluster. The level of the β-hairpin loops is indicated by a dotted line. (**f**) Zoom on the electron density for the DNA at the entrance of the channel together with the model of dsDNA. The residues of the basic cluster are shown and labeled for 2 of the subunits. (**g**) Reconstruction of D5fl at 17 Å resolution without imposed symmetry (Appendix A) with the fitted high-resolution D5_323–785_ structure showing the absence of density for the C-terminal domains. Figure made with Chimera.

**Figure 7 viruses-14-02206-f007:**
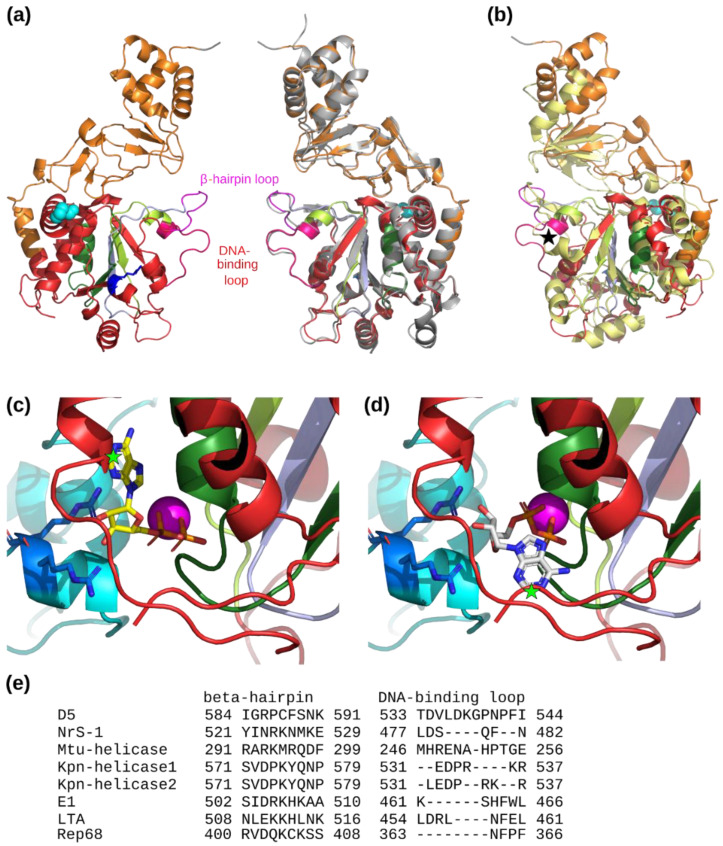
Analysis of the AAA+ helicase and collar domains. (**a**) Two opposite subunits of the refined structure are shown with the AAA+ helicase motifs in color: Walker A, dark green; Walker B, light green; Motif C, light blue; Arg-finger, blue. The residue Pro682 leading to a temperature-sensitive mutant is shown with cyan spheres. The loops corresponding to interactions with DNA in the translocation mechanism are colored in magenta (β-hairpin loop) and bright red (DNA-binding loop). Right subunit: structure from the Alphafold2 prediction (gray) compared to the refined C6 structure. (**b**) Superposition of the NrS-1 AAA+ helicase domain (bright yellow) on the corresponding domain of D5 colored as in panel (**a**). A black star indicates the much shorter DNA-binding loop in NrS-1 helicase. (**c**) View of the cartoon structure around the nucleotide binding site at the subunit interface of D5_323–785_ with the ADP and Mg^2+^ ion positions inferred from papillomavirus E1 (pdb entry 2gxa, chain A). The superposition of the two structures used the AAA+ helicase motif from Iyer and coworkers [12] colored as in panel (**a**). The neighboring chain contributing the arginine finger (blue sticks) is colored in cyan. The predicted clash of the base with the helicase structure is indicated by a green star. (**d**) Same view of D5_323–785_ with ADP and Mg^2+^ ion positions inferred from polyoma virus LTA (pdb entry 1svl, chain A). (**e**) Sequence alignment of the residues forming the β-hairpin and DNA-binding loops of SF3 helicases mentioned in the text where experimental or predicted structural information is available.

**Table 1 viruses-14-02206-t001:** Cryo-EM data collection statistics.

Parameter	Cryo-EM Data Collection on D5_323–785_
Microscope	Technai F30 Polara
High tension (kV)	300
C_s_ (mm)	2
Detector	Gatan K2 Summit (4k × 4k)
Mode	Counting
Energy filter	No
Magnification	42,000×
Calibrated pixel size	1.21 Å
No. of frames per micrograph	40
Nominal defocus range (µm)	−1.5–3.0
Dose par frame (e·Å^−2^)	1
No. of micrographs	830

**Table 2 viruses-14-02206-t002:** Statistics of the refined D5_323–785_ structures and the underlying Alphafold2 model.

	C6	C1	Alphafold2 ^1^
EMDB accession codes	EMD-15574	EMD-15575	
Number of particles used	28425	26710	
Map resolution from Phenix (Å)	3.7	4.6	
FSC resolution ^2^ (Å)	4.1	6.6	
Model composition			
PDB entry	8APL	8APM	
Prot. Res/ Nucleotides	2694/0	2514/16	
Non-hydrogen atoms	21888	20723	
Map correlation within mask	0.73	0.69	
Temperature factor (Å^2^)	207	309	
R.m.s.			
Bond length deviations (Å)	0.002	0.002	0.012
Bond angles (°)	0.476	0.414	1.669
Validation			
Molprobity score	1.94	1.96	0.86
All-atom Clashscore	10.3	13.2	1.33
Rotamer outliers (%)	1.9	2.4	0.24
Ramachandran plot			
Outliers	0.0	0.0	0.0
Favored (%)	96.8	97.8	98.5
Allowed (%)	3.2	2.2	1.5

^1^ Alphafold2 model of residues 323–785. ^2^ Based on a Fourier shell correlation of 0.143.

**Table 3 viruses-14-02206-t003:** Selected similarities of the structural domains of D5 with known structures.

	PDB Entry	Dali Score	rms (Å)	Aligned Residues	out of N Residues	% Sequence Identity
Collar domain						
S17 ribosomal protein	6zxh	6.0	2.7	55	132	7
Papillomavirus E1	2v9p	4.0	3.7	56	269	11
MCM2 (before others)	5v8f	3.9	3.1	64	603	6
Polyomavirus S40 LTA	1svl	3.1	3.4	58	362	7
AAA+ helicase domain						
Primpol-helicase NrS-1	6k9c	16.2	3.2	214	416	17
Papillomavirus E1	5a9k	11.2	3.5	183	269	13
Polyomavirus LTA	4e2i	10.0	4.1	197	362	16
AAV Rep68	7jsi	9.9	4.4	187	276	11
PCV2 Rep	7las	6.3	4.3	130	183	15
C-terminal domain						
Human RFX-DBD	1dp7	4.4	3.1	60	76	10

## Data Availability

The authors declare that the data supporting the findings are available from the corresponding authors upon reasonable request. Coordinates for the two structures have been deposited in the PDB and cryo-EM maps in the EMDB (Table 2).

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
