# Peer review of "The Vaccinia Virus DNA Helicase Structure from Combined Single-Particle Cryo-Electron Microscopy and AlphaFold2 Prediction"

_viruses, 2022, doi:10.3390/v14102206_

Round 1

Reviewer 1 Report

The vaccinia virus DNA helicase structure from combined….

The authors use cryo-electron microscopy with single particle analysis to attain the electron density for a portion of the vaccina virus D5 (323-782). The resolution for the collar domain (323-403) is reported to be 3.7Angstroms, while the resolution for the remaining region (404-785) is reported to be of lower resolution (~6A). The authors use AlphaFold2 to predict the structure of the D5, then fit this model into the electron density to attain a quasi-atomic model of the D5. The work provides the first structure of a helicase for orthopoxviruses and is thus an important piece of work. This is further compounded by the fact that monkeypox virus, which may be the next pandemic to grab a hold of the world, is an orthopoxvirus. Thus, inhibitors of this protein would have therapeutic benefits. I would like to see the following concerns addressed:

1.       In the caption of the images, please indicate what software was used to generate the figures.

2.       The authors need to do a bit more reading and referencing than the BPV E1 and SV40 LTag. There are multiple SF3 viral helicase structures available that include the AAV helicase (Rep68) (Santosh et al., 2020) and the PCV2 helicase (Rep) (Tarasova et al., 2021). Reading these manuscripts and comparing the structures to D5 will strengthen this manuscript.

3.       The mechanism proposed for the BPV E1 is likely to be incorrect. The movement of loops is likely an artifact of crystallography. The ATPase domains of many hexameric AAA+ proteins adopt a spiral staircase (many deposited structures in PDB and EMDB), whereas the loops do not bend as proposed by Enemark et al., 2006). Indeed, there are two hexamers in the ASU of the E1 xtal structure. The hexamer selected and discussed by the Enemark and Joshua-Torr demonstrates movements of this loop; however, the remaining hexamer does not. At the very least, the movement of such loops is controversial, and should not be presented as an accepted model.

4.       Manuscript needs table that describes the cryo-EM information. This tends to be Table 1 in most cryo-EM manuscripts values include: number of micrographs, defocus range, voltage, Cs, detector, number of micrographs, number of particles…

5.       Line 175: Replace “… and allowed finally to obtain…” with “that generated”

6.       There is no evidence that supports the entire molecule to possess C6 symmetry. The collar is likely C6, whereas the ATPase domains may follow C1 symmetry.

7.       Line 216: the word “actually” is not necessary. It can be deleted.

8.       Lines 223-226: Difficult to understand. Please correct grammatical errors.

9.       Table 2: rms for Human RFX-DBD is 31Angstroms. I believe it should be 3.1Angstroms?

10.   Line 232: What is meant by high level?

11.   Line 251: What is meant by “… because of its most homogeneous aspect.”? Is it homogeneous based on Size exclusion, purity, appearance in cryo-EM grid…?

12.   Line 251: The word “actually” is not needed.

13.   Line 257: Please indicate that FATCAT server was used in the caption because Flexible structural alignment is not a routine tool used by the cryo-EM community.  

14.   Lines 265-266: Is there enough space for dsDNA to fit through the pore defined by 6 collar domain? Please provide some measurements for the reader to support the claim of bound dsDNA. How long is the hexamer, how large is the cavity defined by the six ATPase domains, how large is the pore defined by the six collar-domains…?

15.   Figure 5A: The additional density (green arrow) seen in bottom right of 5a is not seen in the middle and bottom left class averages -both of which appear to be side views. Can the authors explain this discrepancy? Perhaps, not all hexamers are bound to DNA

16.   Lines 292-293: Sentence is difficult to understand.

17.   Lines 296-297 (Figure 5): The modeled DNA is difficult to see. Please change colors or increate transparency.

18.   Line 320: Perhaps change “yielded surprisingly” to “surprisingly yielded”

19.   Lines 333:336: This is purely speculatively and worded far too aggressively here. Please either remove the sentence or tone down the argument to indicate that it is speculatory.

20.   Could the authors calculate 3DFSC curves and determine the anisotropy in their dataset?

21.   For the ATPase domains of the cryo-EM figures, please use a threshold where the secondary structures (particularly the alpha-helices) are easily seen. Current figures are contoured at a low threshold and not convincing that they are at the reported resolution.

Author Response

See file.

Reviewer 2 Report

This manuscript is a follow-up of a 2016 study of the poxvirus D5 helicase-primase, which defined the domain organisation of this essential viral protein. Here cryo-EM and alphafold 2 modelling are used to provide convincing models of this elusive structure. This is a significant advance in the field and provides new insights that will undoubtedly underpin rationale elucidation of mechanistics aspects. My concern is the modelling which largely over-interprets the experimental data. With the exception of the collar domain, the model needs to be refer to, and treated as, a model fitted into the EM map, not an experimental structure.

Major:

  • The refinement procedure is not described in any detail. The method section suggests that the two structures were both refined in real-space without additional restraints required for low resolution data. The 6.6 Å structure only warrants rigid-body and shouldn’t be deposited as an experimental structure (or only as C-alphas). The 4.1 Å reconstruction warrants refinement for the good sections (mostly the collar domain) but not the low resolution ones. Refining and depositing such structures is tricky and may require a composite approach with a stated threshold for the domain that are fully refined and the ones that are only modelled by AF2 and rigid body (e.g. C-terminal domain).

  • To allow the reader to evaluate the reliability of the models, the local resolution maps need to be included in the main text. Examples of the typical electron density and model fit for each of the domains would be useful as supplementary material.

  • The manuscript does not provide a satisfactory explanation of what limits the resolution of the C1/C6 reconstructions. The authors should be explicit if this is particle numbers, the quality of the imaging due to technical limitations or flexibility of the protein. If the later is the main hypothesis, focused-reconstructions should be at least attempted.

Minor:

  • L.38: mortality rates are usually quoted as 1% for the West African strain

  • 99/98.3% identical. Where are the differences?

  • L.49: “but which are largely unproven”. This statement needs more details. The drugs have demonstrated good efficacy against disease in non-human primates infected with monkeypox virus but have not been tested in randomised clinical trials.

  • L.55: the authors could make it clear it is their previous study and use “We” instead of “They”.

  • L 61: “bears some resemblance to…”. Explain how so.

  • L. 121: the oligomer seems to be written 3’ to 5’.

  • Fig. 4 is hard to follow. It would help to have annotations of N- and C-termini and better indications of how the missing loops connect. Perhaps a different color scheme or ribbons instead of the cartoon.

  • Fig 5d,f: are these the same views apart from the change in depth of view? It would make more sense to have them as consecutive panels (d,e).

  • Conclusion: “The strong enrichment of the particles observed in the micrographs also speaks in favor of such an absorption (Figure S1a).” What the authors mean by this statement is unclear. Is Fig. S1a supposed to show enrichment at the air-liquid interface? Or does this statement refers to the histograms of orientations?

Author Response

See file.

Reviewer 3 Report

The authors have studied and determined the structure of vaccinia virus D5 helicase using a combination of Cryo-EM and Alpha-Fold2. The authors determined the domain structure, which shed light on understanding the DNA binding site and mechanism of action of D5.

Major Comments:

1. The EMD database ID in table 2 does not show entries in the database. Please share the Cryo-EM maps and corresponding PDB in the supplementary materials for the reviewers if you cannot release them now.

2. The authors determined the Cryo-EM structure, which gave a resolution of 4.1 and 6.6 A in C6 and C1 symmetries. Increasing the number of particles could have improved the classes and, therefore, could have improved resolution close to the DNA binding site. I recommend authors perform Res log analysis and inform if the amount of data/particles collected was sufficient.

3. In Figure 5f, the authors show DNA bound to the tunnel of the hexamer. The fitting appears to be suboptimal. Please show a clear fitting of helices in the slice through electron density figure close to the bound DNA at the top. The residues of the basic cluster shown in blue sticks should be made clear by providing an insight picture of sticks fitting in the density map. 

4. In line 437, the authors say about the limitation of data. Is it the limitation of the resolution captured or the data collected? Please clarify.

5. Some 2D classes show a projection (DNA) from the tunnel, while others do not. It could be some particles have bound DNA, and some do not. Reclassifying the data into bound and unbound DNA and further reconstruction with DNA bound dataset could improve resolution at DNA bound state.

Minor Comments:

1. Write the Cryo-EM data processing details like motion correction in material and methods.

2. Write the details of the construct (D323-785) used to make the Cryo-EM sample complex in material and methods.

3. Write resolution (A) in the figureS1b and d which shows local resolution.

Author Response

See file.

Round 2

Reviewer 3 Report

The authors have addressed the comments but some concerns should be addressed.

1. The authors should upload the validation files of the deposited density map in EMDB database for the reviewers.

2. The authors have addressed the comments in supplementary data but there is no supplementary data uploaded other than one figure.

Author Response

Dear Referee,

1. I apologize for the misunderstanding of your initial request for the validation files. These files are now joined. I have not been really up to date about the current validation procedure with EMDB/pdbe.

2. There has been a mishap in the submission process or the editorial process and you got the graphical abstract instead of the pdf file with the Supplementary material. I join this file and I will submit it again.

I fused the three files in one pdf file, which is joined, containing the information for the C6 structure first, followed by the C1 structure and the Supplementary material, .

Best regards

Wim Burmeister
